# Molecular Simulation Study on Mechanical Properties of Microcapsule-Based Self-Healing Cementitious Materials

**DOI:** 10.3390/polym14030611

**Published:** 2022-02-04

**Authors:** Xianfeng Wang, Wei Xie, Long-yuan Li, Jihua Zhu, Feng Xing

**Affiliations:** 1Guangdong Provincial Key Laboratory of Durability for Marine Civil Engineering, College of Civil and Transportation Engineering, Shenzhen University, Shenzhen 518060, China; xfw@szu.edu.cn (X.W.); xingf@szu.edu.cn (F.X.); 2School of Engineering, University of Plymouth, Plymouth PL4 8AA, UK; long-yuan.li@plymouth.ac.uk

**Keywords:** self-healing, molecular dynamics, microcapsule, mechanical property, cementitious material

## Abstract

Microcapsule-based self-healing concrete can effectively repair micro-cracks in concrete and improve the strength and durability of concrete structures. In this paper, in order to study the effect of epoxy resin on the cement matrix at a microscopic level, molecular dynamics were used to simulate the mechanical and interfacial properties of microcapsule-based self-healing concrete in which uniaxial tension was carried out along the *z*-axis. The radial distribution function, interface binding energy, and hydrogen bonding of the composite were investigated. The results show that the epoxy resin/C-S-H composite has the maximum stress strength when TEPA is used as the curing agent. Furthermore, the interface binding energy between epoxy resin and cement matrix increases with increasing strain before the stress reaches its peak value. The cured epoxy resin can enhance both the interfacial adhesion and the ductility of the composite, which can meet the needs of crack repair of microcapsule-based self-healing cementitious materials.

## 1. Introduction

As a new type of concrete material, microcapsule-based self-healing concrete has been proven to repair micro cracks in concrete and improve the strength and durability of concrete [1]. The repairing principle of microcapsule self-repairing concrete is mainly that when microcracks of concrete extend and contact with the microcapsules, the shell of the microcapsules break, and the epoxy resin diluent flows into the crack and reacts with the curing agent. This fills the cracks and achieves the purpose of repairing the cracks [2,3].

At present, scholars have carried out experimental and simulation studies on microcapsule-based self-healing concrete. Its performance, such as its strength compared with ordinary cement, has been a concern of many scholars, and various studies have been conducted in its experimental aspects [4,5,6,7]. Wang et al. [8] studied the effects of microcapsules on the strength, permeability, and long-term shrinkage of self-healing concrete, and the results showed that the use of concrete containing 10% microcapsules gradually increased its self-healing ability over time. This study showed that self-healing concrete based on microcapsules is feasible and can improve the durability of concrete structures. Sun et al. [9] embedded liquid amine in glycerol tristearate shell to make microcapsule-based self-healing concrete, which successfully repaired cracks in the concrete and improved the porosity of the concrete. Perez et al. [10] mixed amino functionalized silica particles and epoxy resin wrapped by silica into cement material to make a self-healing system. The results showed the stability of epoxy resin in microcapsules and the existence of amino groups bound by functionalized silica particles and silicon atoms in cement. Through the copolymerization of 2-(dimethylamino) ethyl methacrylate and acryloxyethyl trimethyl ammonium chloride, Wang et al. [11] successfully synthesized pH-sensitive hyperabsorbent polymer and realized self-healing for the cement micro-cracks. Through triaxial compression test and mercury injection test, Han et al. [12] studied the influence of different factors such as microcapsule content and preloading stress ratio on mechanical properties and microstructure of self-healing microcapsule concrete. The results showed that the healing effect of cement material was enhanced with the increase in the content of microcapsules. In the aspect of computer simulation, Wang et al. [13] established a numerical model of microcapsules by combining the finite element method with the MATLAB software, and studied the precise composition parameters of the urea-formaldehyde resin microcapsule model. After previous experiments and simulation studies on microcapsule-based self-healing concrete by scholars, it has been proved that it can effectively improve the strength and durability of concrete. However, the microscopic level of understanding has been lacking in previous studies. The simulation of microcapsule-based self-healing concrete at the micro level can be used to study its mechanical properties at a deeper level to reduce the experimental error and waste of time.

In recent years, molecular dynamics (MD) simulation has been widely used in the microscopic study of cement-based materials [14,15,16,17,18]. MD is a method that uses classical mechanics to solve the motion behavior of atoms and uses statistical mechanics to calculate the properties of various systems. Molecular dynamics, as a powerful tool, can be well used to study interface relations [19]. Many scholars have used it to study the interface relations between epoxy resin and other materials [20,21,22]. In the study of Du et al. [20], a composite model of epoxy resin and C-S-H was established, and MD simulation results showed that electrostatic interaction of calcium ions in C-S-H was the main source of binding energy between epoxy resin and C-S-H layer. Hou et al. [23] found that water molecules weaken the interaction energy between epoxy resin and C-S-H by breaking the hydrogen bond and the Ca-O bond, thus reducing the energy required for interface stripping and shear failure. To sum up, molecular dynamics can be used to conduct mechanical research on microcapsule-based self-healing concrete,

The main factors influencing the repair effect of microcapsule-based self-healing concrete are as follows: (1) curing reaction of repair agent; (2) permeability of repair agent in crack; (3) destruction patterns of microcapsules; and (4) the bonding strength of the remediation agent on the surface of the cement matrix. In the authors’ previous studies on microcapsule-based self-healing concrete using molecular dynamics [24,25], the interaction between the microencapsulated shell (urea-formaldehyde resin) and core (epoxy resin) and C-S-H has been studied. The results showed that the microcapsule shell and core have good interfacial bonding energy with the surface of cement matrix, which can meet the requirements of repairing cracks.

However, there is still a lack of research on the interface behavior between epoxy resin and C-S-H composites under tensile process. In this study, molecular dynamics simulation was used to study the mechanical properties of the interface between C-S-H and the microcapsule core (epoxy resin). Since the curing of epoxy resin is one of the important reasons affecting the effect of fracture repair, the purpose of this paper is to study the influence of two different epoxy curing agents on the bonding energy of epoxy resin and cement matrix during tensile process. Three kinds of C-S-H/epoxy resin models were established in this paper: one was the epoxy resin model without curing, and the others were the epoxy resin models using MC120D and tetrethylenepentamine (TEPA) as curing agents, respectively. The interfacial properties of the composite model were studied by uniaxial tensile simulation in the Z-direction. Finally, the simulation results were analyzed by the stress-strain curve, interface binding energy, and radial distribution function.

## 2. Computational Methodology

### 2.1. Model Construction

According to Zhang’s experiment study [26], E-51 epoxy resin was used as the core of the microcapsules, TEPA and MC120D were used as the curing agents, and N-butyl glycidyl ether (BGE) was used as the diluent. Epoxy, TEPA, MC120D, and BGE were modeled using Materials Studio as shown in Figure 1, Figure 2, Figure 3 and Figure 4. After the four models were established, the cured epoxy resin model, as shown in Figure 5 and Figure 6, was established according to the curing principle [27]. In this curing reaction, the ratio of epoxy resin to BGE epoxy group was 2:1, which was from Zhang’s experiment [26]. Tobermorite 14 Å was built to simulate the C-S-H, which has been proved to have an interlayer structure similar to C-S-H and many scholars have used tobermorite to simulate C-S-H by molecular dynamics simulation [28,29,30]. Tobermorite 14 Å shown in Figure 7 is used in this study. Geometry optimization was first implemented to minimize the energy of tobermorite 14 Å in order to optimize and relax the structure. The geometry optimization program in the Forcite module was used to optimize the geometric structure. The optimization algorithm is the smart algorithm (a cascade of methods using successively steepest descent, ABNR, and quasi-Newton methods). When the energy of the model reaches minimum and keeps stable, it can be considered that the model has been optimized [15]. For the purpose of building layer structure, the XY plane of the optimized tobermorite 14 Å was determined. Then the whole model was built as shown in Figure 8, in which the two sides are tobermorite 14 Å and the middle layer is an epoxy resin model. Meanwhile, the atomic numbers of the three models are 7177 (n = 0), 7240 (MC120D), 7236 (TEPA) respectively. The dimensions of the three models are 62 Å ∗ 20 Å ∗ 65 Å (n = 0), 62 Å ∗ 20 Å ∗ 68 Å (MC120D) and 62 Å ∗ 20 Å ∗ 65 Å (TEPA), respectively. After establishing the entire composite model as described above, geometry optimization was implemented again to achieve a stable conformation. And, the velocity Verlet algorithm and periodic boundary conditions is used throughout the simulation process. Last, the model was calculated by using NPT ensemble (N is the number of atoms, P is the pressure, T is the temperature) 1000 ps and NVT ensemble (N is the number of atoms, V is the volume, T is the temperature) for 2000 ps in turn using the COMPASS forcite. COMPASS is an ab initio force field whose structure is largely inherited from an earlier force field known as CFF [31,32]. Most parameters are derived from ab initio data. Previous studies by the researchers have shown that the COMPASS force field can well predict the properties of cementitious materials [33,34,35].

### 2.2. Uniaxial Tension Simulation

Uniaxial tensile simulation of the model in the Z direction was carried out to obtain the stress strength and failure strain under the NPT ensemble condition. The specific steps were as follows: (1) the temperature was set as 300 K, the external pressure was set as 0 GPa and the calculation time was 1 ns; (2) after the stress relaxation in each direction is zero, input the set stress parameters and set the stretching step of the model in the Z direction as 1 fs in the simulation process. In order to analyze the simulation results, the calculation results were saved every 10,000 steps.

## 3. Results and Discussion

### 3.1. Stress-Strain Curve

The stress-strain curve obtained by uniaxial tensile simulation is shown in Figure 9, and the failure strain and peak stress are shown in Table 1. As with the stress-strain curves of tobermorite or tobermorite/urea-formaldehyde resin obtained by other scholars [24,36], all three stress-strain curves have three stages, namely, the elastic stage (I), the yield stage (II), and the failure stage (III). As can be seen from the Figure 9, the three models are basically the same in the elastic stage, while differences begin to appear in the yield stage. The failure strains of the three models n = 0, MC120D and TEPA are 0.17, 0.30, and 0.28, respectively. Compared with the results of 0.2 pure tobermorite model [36], the failure strain of the cured epoxy resin-cement matrix model increased, while the failure strain of the uncured model decreased slightly, indicating that the presence or absence of curing had a greater effect on the strain of the microcapsule-based self-healing concrete. The stress strength of the three models n = 0, MC120D and TEPA were 2.52 GPa, 2.62 GPa, and 2.80 GPa, respectively. Compared with the results of pure tobermorite model in the literature (the stress strength were 0.87 GPa, 0.93 GPa, and 1.01 GPa) [36], the stress strength of all three models increased, indicating that the addition of epoxy resin can improve the strength of cement matrix and achieve the purpose of repairing cracks. At the same time, the strength of the cured epoxy resin is greater than that of the uncured one, indicating whether epoxy resin is cured or not has a certain influence on the strength of cement-based materials. In conclusion, in the experiment, the proportion and uniform dispersion of curing agent should be controlled to ensure that the epoxy resin in the crack is cured in time, so as to improve the strength of cement-based materials. In addition, TEPA can be used as the curing agent to improve stress strength at room temperature, which is consistent with the results of Zhang’s experimental research [26]. From the analysis in our previous research [25], we can also know that using TEPA as the curing agent has better interface bonding force and atomic interaction between interfaces, which is also the reason why the stress strength of the model using TEPA as the curing agent increases.

### 3.2. Mechanical Properties

The mechanical properties of the models were calculated after structural optimization, and the maximum strain is set as 0.003 to ensure the model is within the elastic stage. In order to ensure the accuracy of the calculation results, the module was repeated three times for each model, and the two close values were averaged to obtain the final results. The bulk modulus, shear modulus, and elastic modulus of the three composite models in X, Y, and Z directions are shown in Table 2.

Based on the bulk modulus and shear modulus calculated, the Poisson’s ratio ν and Young’s modulus E of the whole model were obtained according to Equations (1) and (2) in which the structures were approximately treated as isotropic. The calculated results are shown in Table 3.
(1)$$\nu =\frac{3-2G/K}{6+2G/K}$$
(2)$$E=\frac{9G}{3+K/G}$$

According to the molecular dynamics and experimental results of Du et al. [20], the Young’s modulus of epoxy resin and C-S-H composites were 28 GPa and 9.8–35 GPa, respectively. In other references, the elastic modulus of microcapsule-based self-healing concrete is between 15–45 GPa [37,38]. The elastic modulus obtained by the simulation in this paper is slightly higher than that obtained in other references, mainly for the following reasons: (1) in the molecular dynamics simulation of Du et al. [19], the C-S-H model was built according to the Taylor’s hypothesis, and the epoxy resin layer was composed of multiple epoxy resin chains. In our research, the tobermorite model was established to simulate C-S-H, and the epoxy resin only built one chain, so that makes a difference; (2) the microcapsule-based self-healing concrete includes complete microcapsules (including urea resin and epoxy), and the concrete contains not only C-S-H, but also ettringite, silicate and other hydration products, which also has a certain influence on the simulation results; and (3) the test results of elastic modulus are also affected by porosity [39,40].

In the actual microcapsule-based self-healing concrete, the epoxy resin is not just a chain, but has a certain proportion in concrete. In order to study the influence of different volume ratios of epoxy resin in the cement matrix on the elastic modulus of the whole model and obtain the elastic modulus closer to the actual situation, the shear modulus and elastic modulus of the composite models with different epoxy resin content were calculated using the Mori-Tanaka method (MT method) [41], which is used for calculating the average modulus of a composite containing inclusions. This calculation method takes into account the interaction between inclusions in composite materials and is widely used to solve the properties of heterogeneous composite materials. According to this method, when the composite material is composed of two different components, the bulk modulus and shear modulus of the composite are calculated by Equations (3) and (4).
(3)$$K_{ij}=\frac{4K_{i}G_{i}C_{j}+3K_{i}K_{j}+4K_{j}G_{i}C_{i}}{4G_{i}+3K_{j}C_{j}+3K_{i}C_{i}}$$
(4)$$G_{ij}=G_{i}\frac{6G_{j}\left(K_{i}+2G_{i}\right)+\left(9K_{i}+8G_{i}\right)\left(G_{j}C_{i}+G_{i}C_{j}\right)}{6\left(K_{i}+2G_{i}\right)\left(G_{i}C_{i}+G_{j}C_{j}\right)+9K_{i}G_{i}+8G_{i}^{2}}$$
where, $K_{ij}$ and $G_{ij}$ represent the bulk modulus and shear modulus of composites, respectively.$\;K_{i\left(j\right)}$ and $G_{i\left(j\right)}\;$represent the bulk modulus and shear modulus of the composites, respectively. $C_{i}$ and $C_{j}$ represent the proportion of the two kinds of crystals *i* and *j* in the composite material, respectively. According to Equations (3) and (4), the shear modulus and volume modulus of composites with different epoxy resin volume fractions were calculated, and then Young’s modulus was calculated from Equation (2). The results are shown in Table 4 and Figure 10. It can be seen from the table that the Young’s modulus of the three composites is between 17 and 50 GPa when the epoxy resin is in the range of 1% to 30%, which is similar to the experimental results of 15–45 GPa [37,38]. It can be seen from the figure that the Young’s modulus of the three composites decreases with the increase of epoxy resin content. In the simulation study of Du [20], it is also found that the Young’s modulus decreases with the increase of resin content, since more resins can offset more loads through plastic deformation [42].

It can be seen from Table 3 that the elastic moduli of the three models are 54.14 GPa, 48.26 GPa, and 55.41 GPa, respectively, while the curve in Figure 10 can be fitted to show that the volume fractions of epoxy resin chain in the three models of n = 0, MC120D and TEPA are 0.43%, 0.39%, and 1.3%, respectively. According to the volume and calculation of atoms and covalent bonds of the epoxy resin chain in the model, the volume content of epoxy resin in the three models is 0.24%, 0.53%, and 0.49%, respectively. According to the comparison between the two sets of data obtained by mechanical property calculation in MS and MT method, there are certain differences for the following reasons: (1) when calculate in simulation, the elastic modulus is obtained at a certain tensile rate, while the MT method only calculates the elastic modulus of static model, which will cause a certain gap; (2) in the MD simulation, in addition to the bonding effect, the model will also have influence of non-bonding effects, such as the van der Waals and electrostatic effects. However, under the MT method, the epoxy resin chain can only calculate the bonding volume, which will lead to certain errors.

### 3.3. Binding Energy

The interfacial binding energy can directly reflect the interaction between the tobermorite and epoxy layers, therefore the change of interfacial binding energy during the complete tensile process was studied. The binding energy changes of the three models in the uniaxial stretching process were obtained from Equation (5) [43], which is for calculating the interface binding energy, and are shown in Figure 11. Since the interface area between the tobermorite and epoxy resin may vary during the tensile process, the interface binding energy per unit area can be calculated by Equation (6) as follows:(5)$$E_{b}=-E_{I}=E_{\mathrm{total}}-\left(E_{\mathrm{tobermorite}}+E_{\mathrm{epoxy}}\right)$$
(6)$$E=\frac{E_{b}}{A}$$
where $E_{b}$ is the binding energy between tobermorite and epoxy resin, $E_{I}$ is the interaction energy between tobermorite and epoxy resin, $E_{\mathrm{total}}$ is the total energy of the whole model, $E_{\mathrm{tobermorite}}$ and $E_{\mathrm{epoxy}}$ are, respectively, the energy of the tobermorite and epoxy resin, E refers to the interfacial binding energy per unit area and A refers to the interfacial area between tobermorite and epoxy resin. The calculated interface binding energy per unit area is shown in Figure 12.

It can be directly seen from Figure 11 and Figure 12 that, in the elastic stage of the stretching process, the interface binding energy of the three models generally increases with the increase of strain. This indicates that the interaction between the tobermorite and epoxy resin increases gradually during the tensile process, thus slowing down the rate of interface failure, which also explains why the stress strength of epoxy resin added is greater than that of pure tobermorite. However, when the strain exceeds the failure strain, the interface binding energy decreases rapidly, which is due to the interface failure, resulting in the decrease of the interaction between the interfaces. After the addition of epoxy resin, although the ductility of the material can be improved to a certain extent, the material still presents a state of brittle failure after exceeding the ultimate strain.

By comparing the strain energy of the three models in the tensile process, it can be seen that the interface binding energy of the epoxy resin-tobermorite model with the curing agent increases over 50% compared with that of the uncured epoxy resin-tobermorite model, and the use of TEPA as curing agent can better increase the interface binding energy. Thus, the strength of the whole model is increased, which corresponds to the result of the stress-strain curve. In general, after curing, epoxy resin can increase the bonding effect between epoxy resin and cement matrix, improve the strength of concrete, and thus delay the interface failure.

### 3.4. Radial Distribution Function

Radial distribution function (RDF) is a characterization method for calculating the adsorption points on composite surfaces of two materials and is used to describe the spatial correlation between atoms [44]. The RDF ($g_{AB}\left(r\right)$) represents the ratio of the probability of finding B atoms in a spherical shell with a distance r from the center of A atoms and a thickness of δr to the probability that B atoms are uniformly distributed in the entire simulation system. The calculation formula is as follows:(7)$$g_{AB}\left(r\right)=\frac{dN}{\rho 4\pi r^{2}\delta r}$$
where, dN represents the number of B atoms in the range r to δr away from A atoms, and ρ is the average density of the whole model. The RDF is used to understand the molecular structure of the model, especially the strength of the interaction between the two types of atoms. The position and sharpness of the peaks in the radial distribution function reflect the degree of structural order between the two types of atoms, and researchers had used RDF to analyze the strength of the interaction between the two atoms [22,45].

The radial distribution function was chosen to analyze the atomic interaction between the two layers of tobermorite and epoxy resin in the tensile process. The conformation with the highest stress among the three models was selected for analysis, and the results are shown in Figure 13. Conforming with the previous analysis [25], the interaction between tobermorite and epoxy resin is mainly due to the fact that Ca ions in tobermorite and O and N atoms in epoxy resin, while nitrogen atoms do not exist in the uncured epoxy resin. Therefore, the interaction between atoms in the n = 0 model is weaker than the other two models. This also leads to a smaller interfacial binding energy as mentioned in Section 3.3. In addition, the positions of the first peaks of RDF of Ca and O atoms in n = 0, MC120D and TEPA models are 2.23 Å, 2.19 Å, and 2.19 Å, respectively. It can be seen that the position of the first peak in the cured models is more forward than that in the uncured one, indicating that the interaction in these two models is stronger. Thus, the cured epoxy resin can enhance interaction with tobermorite. As can be seen from Figure 13b,c, the cured model increases the interaction between Ca ions and N atoms. It is obvious that when TEPA is used as curing agent, the peak at the first position of RDF of Ca and N is more forward and the interaction is greater. In conclusion, the N atoms in TEPA can enhance the interaction between the epoxy and the cement matrix, such that we can choose a more suitable curing agent in the experiment.

### 3.5. Hydrogen Bonding

Another important interaction between the tobermorite layer and epoxy layer is hydrogen bonding, in addition to the main atoms mentioned in Section 3.4. When the first peak distance of RDF between the O atom and H atom is less than 2.45 Å, a hydrogen bond is formed between them [46]. Therefore, Figure 14 shows the RDF diagrams of H atoms in tobermorite and O atoms in epoxy resin for the three models at peak stress. As can be seen, the first peak of RDF in the uncured epoxy resin model is 2.49 Å, which is more than 2.45 Å, so there is no hydrogen bond between tobermorite and epoxy resin. However, the first peak of the model using MC120D and TEPA as curing agent was 1.69 Å and 1.59 Å, respectively, both of which were less than 2.45 Å, forming a hydrogen bond. Between them, the TEPA model is more forward, indicating that its hydrogen bonding is stronger, thus increasing the interaction between the tobermorite and epoxy layers. Compared with the previous studies [25], when the model is stretched and in the conformation with the peak stress, the RDF peak position is more forward and sharp, indicating that during the stretching process of the model (and before the stress intensity reaches its peak), the hydrogen bonding is strengthened, which can slow down the interface damage and thus increase the strength of the model.

## 4. Conclusions

In order to study the effect of epoxy resin on the strength of concrete, in this paper, through establishing three models, namely the non-curing model and the epoxy resin/C-S-H composite models using MC120D and TEPA as curing agents, uniaxial tensile simulation of the models along the *z*-axis was carried out. The following conclusions can be drawn:(1)The stress-strain curves proved that using a curing agent can increase the strength of the model, in which the highest strength was from TEPA.(2)The stress in the epoxy resin/C-S-H composite increases due to the increase of interface binding energy in the two stages before failure in the tensile process.(3)The reason why the model using TEPA as curing agent has the greatest strength should be attributed to the greater interaction between O and N atoms in the epoxy resin cured by TEPA and Ca ions in tobermorite, as well as stronger hydrogen bonding between tobermorite and epoxy resin.(4)The tensile simulation in this paper proves that the cured epoxy resin can enhance both the ductility and the strength of concrete, so that it can achieve self-healing function of concrete.(5)In this study, we mainly focus on studying the influence of curing agents on the repairing effect. In fact, there are still other factors that have not been considered (i.e., various amine compounds with a different number of amino groups), which should be studied in future.

## Figures and Tables

**Figure 1 polymers-14-00611-f001:**
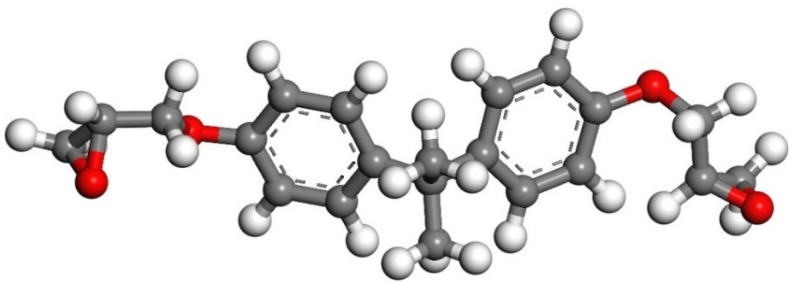
The model of E-51. (n = 0; Color legend: hydrogen H (white); oxygen O (red); carbon C (grey)).

**Figure 2 polymers-14-00611-f002:**
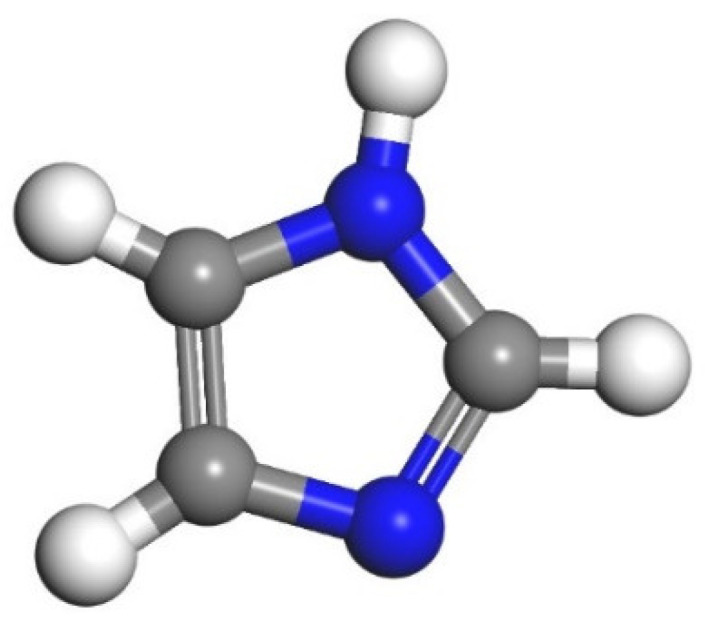
The model of MC120D. (Color legend: hydrogen H (white); nitrogen N (blue); carbon C (grey)).

**Figure 3 polymers-14-00611-f003:**
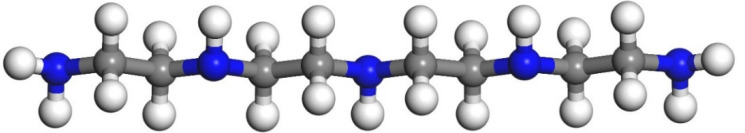
The model of TEPA. (Color legend: hydrogen H (white); nitrogen N (blue); carbon C (grey)).

**Figure 4 polymers-14-00611-f004:**
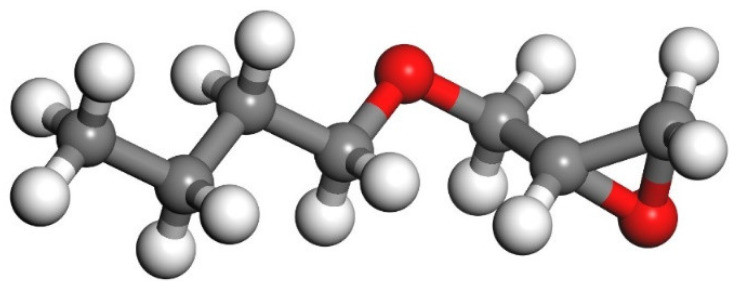
The model of BGE. (Color legend: hydrogen H (white); oxygen O (red); carbon C (grey)).

**Figure 5 polymers-14-00611-f005:**
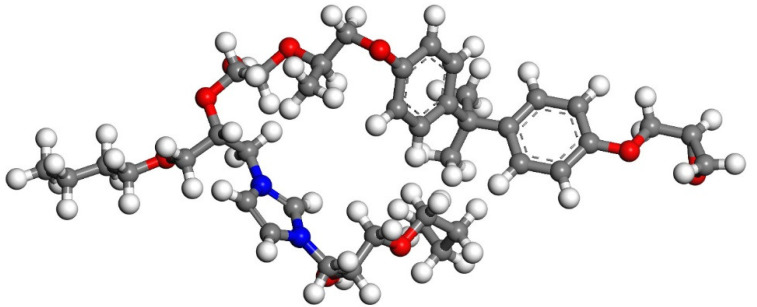
The model of epoxy resin cured with MC120D. (Color legend: hydrogen H (white); oxygen O (red); carbon C (grey); nitrogen N (blue)).

**Figure 6 polymers-14-00611-f006:**
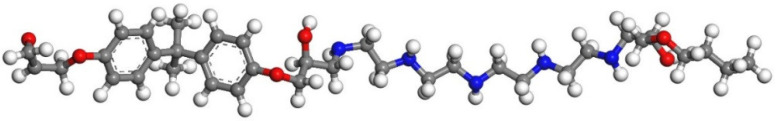
The model of epoxy resin cured with TEPA. (Color legend: hydrogen H (white); oxygen O (red); carbon C (grey); nitrogen N (blue)).

**Figure 7 polymers-14-00611-f007:**
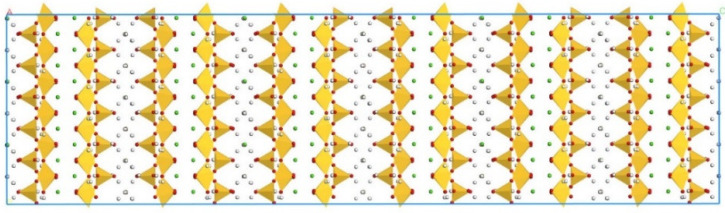
Molecular structure of tobermorite 14 Å. (Color legend: hydrogen H (white); calcium Ca (green); oxygen O (red); silica Si (yellow polyhedral)).

**Figure 8 polymers-14-00611-f008:**
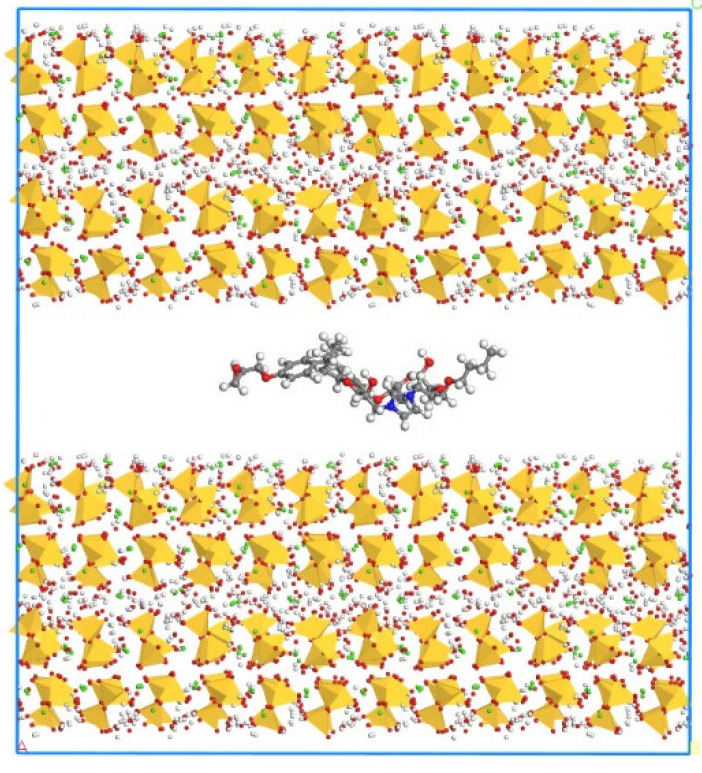
The whole model of tobermorite/epoxy. (Color legend: hydrogen H (white); calcium Ca (green); oxygen O (red); silica Si (yellow polyhedral)).

**Figure 9 polymers-14-00611-f009:**
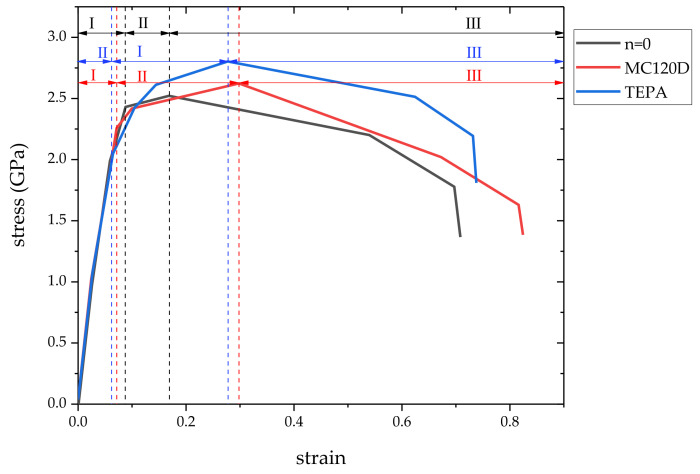
Stress–strain curves. (note: I, elastic stage; II, yield stage; III, failure stage).

**Figure 10 polymers-14-00611-f010:**
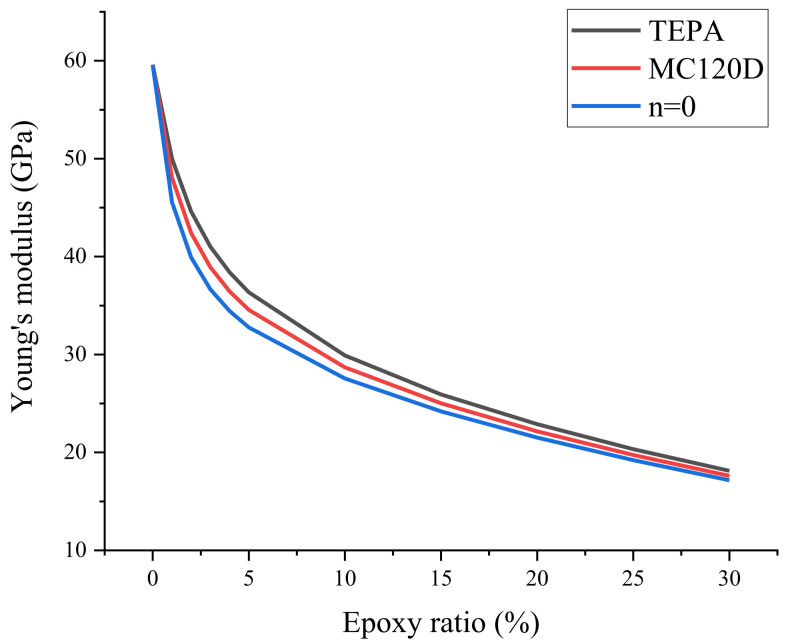
Young’s modulus of different epoxy resin volume proportions.

**Figure 11 polymers-14-00611-f011:**
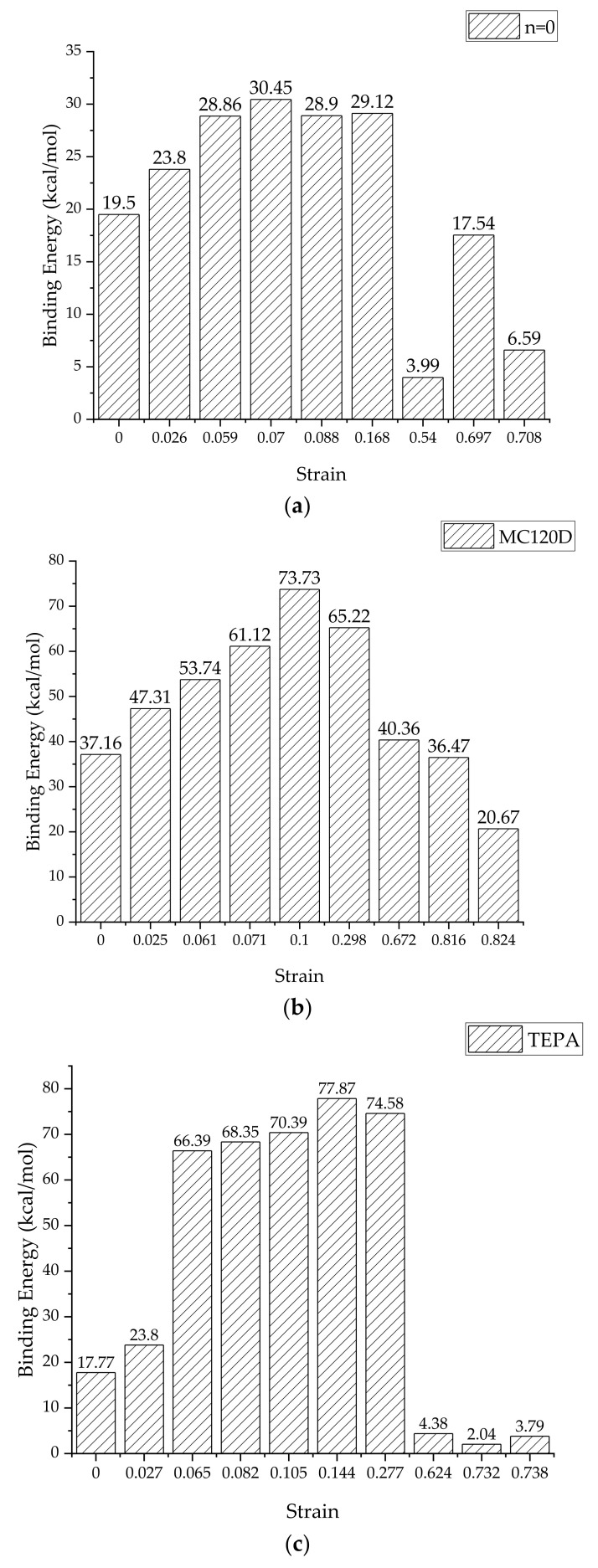
The binding energy of three models (**a**) n = 0, (**b**) MC120D, and (**c**) TEPA.

**Figure 12 polymers-14-00611-f012:**
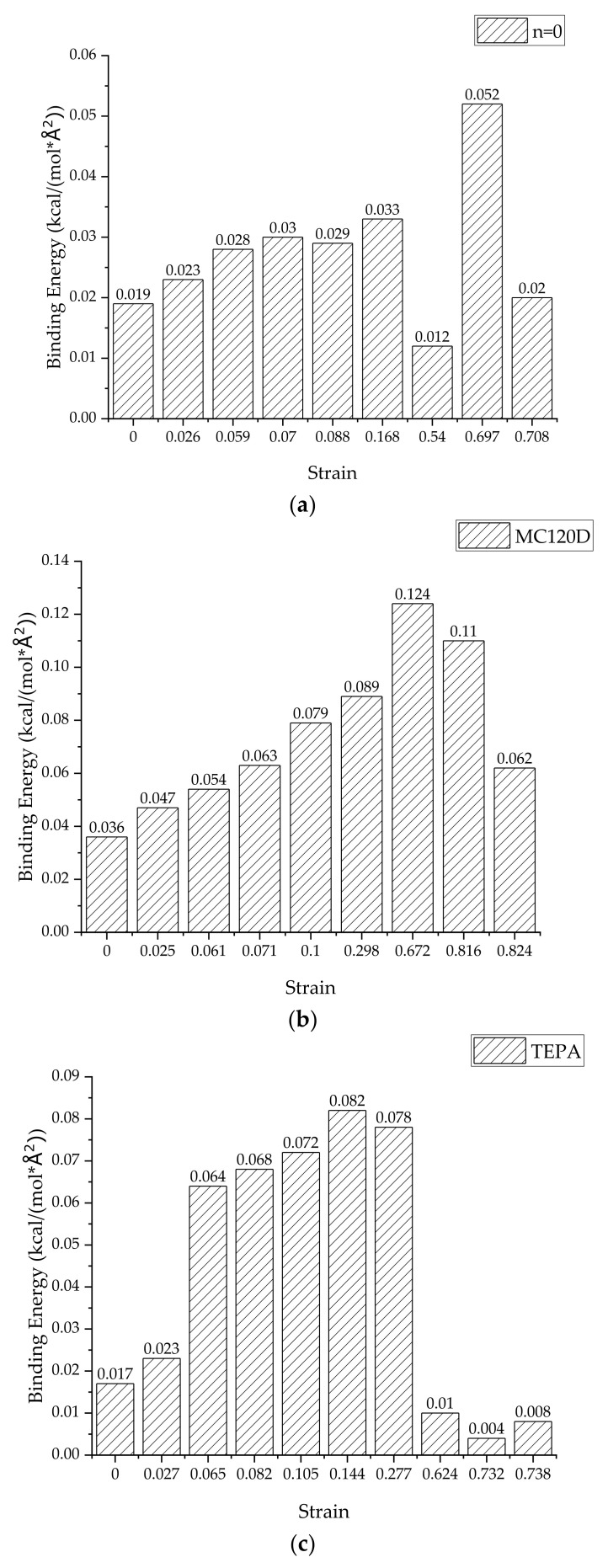
Interfacial binding energy per unit area of three models (**a**) n = 0, (**b**) MC120D, and (**c**) TEPA.

**Figure 13 polymers-14-00611-f013:**
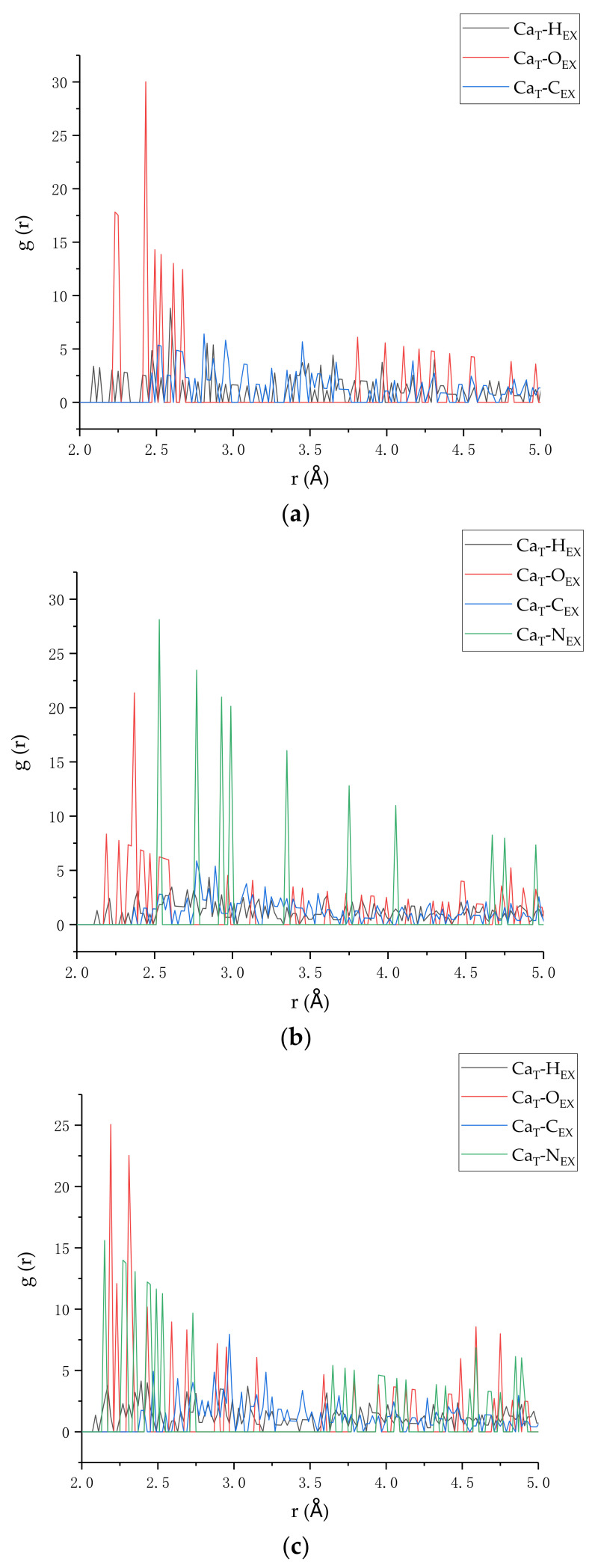
The RDF of three models (**a**) n = 0, (**b**) MC120D, and (**c**) TEPA.

**Figure 14 polymers-14-00611-f014:**
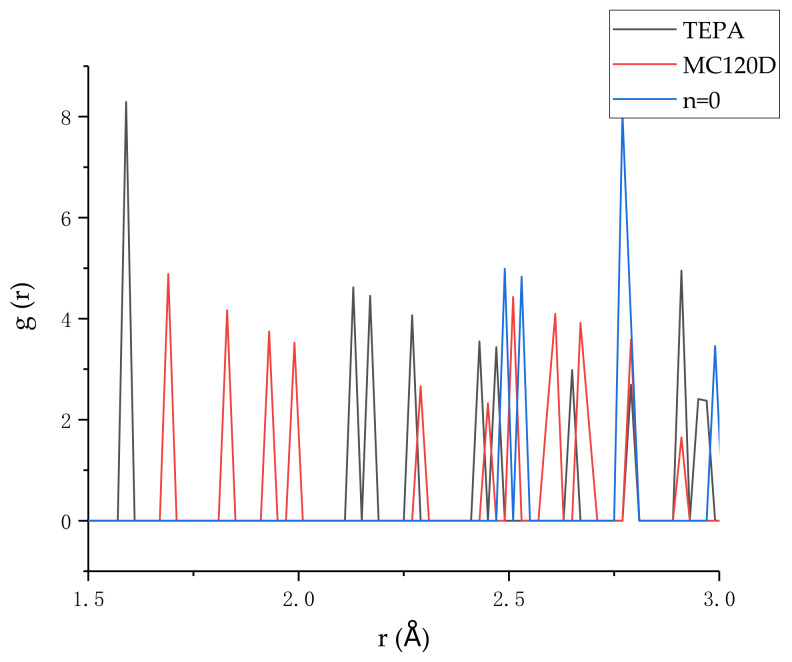
The RDF of H in tobermorite and O in epoxy.

**Table 1 polymers-14-00611-t001:** Failure strain and tensile strength.

Model	Failure Strain	Tensile Strength (GPa)
n = 0	0.17	2.52
MC120D	0.30	2.62
TEPA	0.28	2.80

**Table 2 polymers-14-00611-t002:** The bulk modulus, shear modulus and Young’s modulus of the three models (GPa).

Model	Bulk Modulus (Variance) (K)	Shear Modulus (Variance) (G)	Young’s Modulus (Variance) (E)		
			X	Y	Z
n = 0	48.33 (0.11)	20.61 (0.06)	54.76 (2.07)	70.24 (0.41)	55.89 (2.34)
MC120D	46.56 (0.00)	18.18 (0.01)	49.45 (0.00)	64.69 (0.01)	49.84 (0.00)
TEPA	47.89 (0.33)	21.19 (1.12)	53.35 (0.72)	70.40 (1.20)	50.74 (0.83)

**Table 3 polymers-14-00611-t003:** Poisson’s ratio and Young’s modulus of the three models.

Model	Poisson’s Ratio	Young’s Modulus (GPa)
n = 0	0.31	54.14
MC120D	0.33	48.26
TEPA	0.31	55.41

**Table 4 polymers-14-00611-t004:** Young’s modulus of different epoxy resin volume proportions.

Epoxy Ratio (%)	Young’s Modulus (GPa)		
	n = 0	MC120D	TEPA
0	59.59	59.59	59.59
1	45.57	48.05	50.00
2	39.91	42.41	44.59
3	36.66	38.91	41.01
4	34.44	36.45	38.39
5	32.77	34.56	36.34
10	27.56	28.69	29.91
15	24.20	25.02	25.93
20	21.52	22.16	22.88
25	19.21	19.74	20.34
30	17.16	17.61	18.12

## Data Availability

Data available on request.

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
