# Peer review of "Molecular Simulation Study on Mechanical Properties of Microcapsule-Based Self-Healing Cementitious Materials"

_polymers, 2022, doi:10.3390/polym14030611_

Round 1
Reviewer 1 Report
This manuscript written by Wang et al. presented computational results focused on a very practical hot-topic, namely self-healing cement. Indeed, self-healing cement is a beautiful research topic for biomedical materials, civil engineering, oil&gas, and many others. You may find many papers and even more patents for this topic. The authors showed many interesting findings, but their conclusions and discussion lack the link to experimental results and are based only on the COMPASS forcite results, i.e., MD and ab initio calculations. I think that some improvements are required to enhance the quality of this paper:
- First of all, please justify the reliability of your model highlight the choice of COMPASS and the process of the geometry optimization
- Secondly, why you omit most important publications dealing with the self-healing cement showing the experimental results: These references are missing:
https://doi.org/10.1016/j.matchemphys.2015.08.047
https://doi.org/10.1016/j.cemconcomp.2018.11.023
https://doi.org/10.1016/j.cemconcomp.2021.104132
- The choice of TEPA and its efficiency is quite understandable, but everybody already used TEPA and patented this particular amine compound for self-healing cement. Here I would expect that you may simulate various amine compounds with a different number of amino groups, different lengths of the aliphatic chain, etc. Why you did not do that? Can you extend your research to a higher number of compounds? Otherwise, I dont see the value of this particular work?!
Author Response
Please kindly find the attached word file: response to reviewers.

Reviewer 2 Report
The manuscript investigates potential concrete self–repairing by
embedding micro–capsules containing epoxy resin diluents and curing
agents. These are expected to react as micro-cracks set inside the
concrete. To this end, some important physicochemical properties of
three models based on epoxy resin – one pure and two containing
different curing agents (MC120D, an imidazole type hardener, and
tetrethylenepentamine) were explored by molecular dynamics. It was
found that while both hardeners may be used for producing
self–repairing concrete, tetrethylenepentamine is superior to MC120D
since it provides better mechanical properties to the epoxy resin
system.
The results may be of interest to scientists and engineers working in
the field of cement production. Unfortunately the presentation is poor
and the text is not easily accessible to interested readers. In my
opinion it needs extensive editing before it can be considered for
publication. The following major point may help the authors to improve
the quality of their manuscript.
– How many molecules are included in the simulation? The number of
particles is crucial to any molecular dynamics study.
– It is not clear what concentrations of the healing agents are
present in the epoxy resin system and the diluent.
– The computational procedure is not well described: Which method is
used to integrate the equations of motion? How large is the simulation
box? Which boundary conditions are employed?
– The radial distribution function usually provides information on the
structure of the system under consideration. It cannot be used
directly to describe the interaction among the different constituents.
I urge the authors to elaborate on how they were able to draw their
conclusions on the interaction between the distinct atoms.
Author Response

(The authors gave the same response as above.)

Round 2
Reviewer 1 Report
Authors answered all questions that were addressed.
Reviewer 2 Report
The authors have improved the quality of the presentation. Thus recommend publication of the manuscript.